# Few-shot Image Generation with Elastic Weight Consolidation

**Yijun Li**    **Richard Zhang**    **Jingwan Lu**    **Eli Shechtman**

Adobe Research
{yijli, rizhang, jlu, elishe}@adobe.com
https://yijunmaverick.github.io/publications/ewc/

## Abstract

Few-shot image generation seeks to generate more data of a given domain, with only few available training examples. As it is unreasonable to expect to fully infer the distribution from just a few observations (e.g., emojis), we seek to leverage a large, related source domain as pretraining (e.g., human faces). Thus, we wish to preserve the *diversity* of the source domain, while adapting to the *appearance* of the target. We adapt a pretrained model, without introducing any additional parameters, to the few examples of the target domain. Crucially, we regularize the changes of the weights during this adaptation, in order to best preserve the "information" of the source dataset, while fitting the target. We demonstrate the effectiveness of our algorithm by generating high-quality results of different target domains, including those with extremely few examples (e.g., $\leq 10$). We also analyze the performance of our method with respect to some important factors, such as the number of examples and the dissimilarity between the source and target domain.

## 1   Introduction

The success of generative adversarial networks (GANs) [8] has typically been illustrated with large amounts of training data, for example, 70,000 images for just a specific domain (aligned faces) [16] or 1.3M images across different classes [34]. However, many practical use cases provide limited data. For example, in the artistic domain, it is at best cumbersome, and at times prohibitive, to hire artists to make thousands of creations. While generative models currently struggle in this low-data regime, our goal is to generalize from a few, new examples.

A key component to this is the ability to leverage prior experience. For example, we can use our knowledge of variations in the appearance of natural faces to easily imagine variations of a specific, given cartoon face. In this work, we aim to give generative models the same ability, as shown in Figure 1. More formally, we study the problem of few-shot image generation in a continuous learning framework – training an algorithm to generate more data of a target domain, given only a few examples. An underlying assumption with this setup is that the source and target domains share some latent factors, with some differences related to their distinct difference in appearance. For example, when transferring from real natural faces to emojis, variations in pose and expression can be naturally extended to the target domain.

To achieve this goal, we propose a straightforward and effective adaptation technique. That is, we adapt the pretrained model's weights, without introducing additional parameters. Fixing the architecture implies that tedious manual designs on new parameters (e.g., number of parameters, their position, etc.) are not necessary. Instead, the challenge is how to adapt the weights to fit the appearance of the limited target domain data, while retaining as much transferred knowledge, or *diversity* from the source.

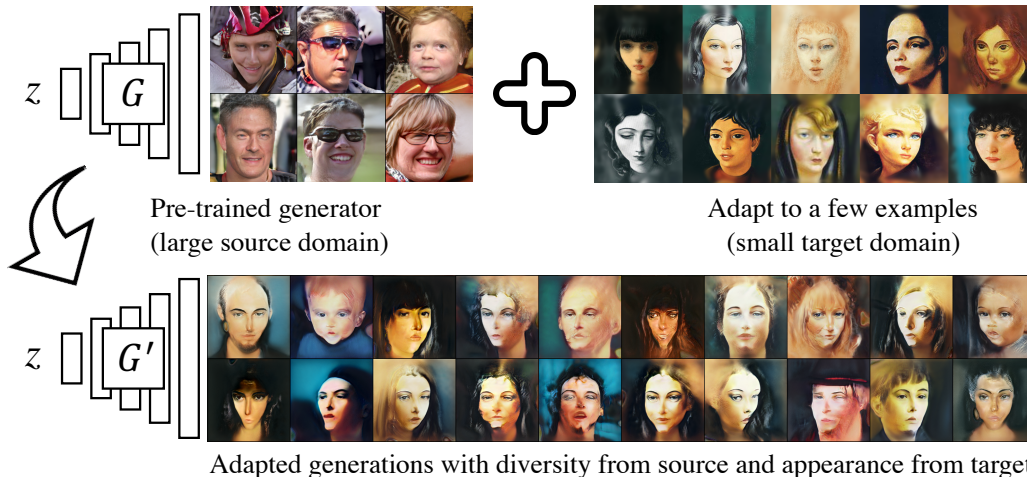

Pre-trained generator
(large source domain)

Adapt to a few examples
(small target domain)

Adapted generations with diversity from source and appearance from target

Figure 1: Pipeline of our few-shot image generation. We first pretrain a generative model on the source domain (e.g., real faces) with a lot of data. We then adapt it to the target domain (e.g., Moïse Kisling faces [44]) with just a few examples to generate more data in target domain (all images are of size 256×256).

A key property to note is that weights have different levels of importance; thus, each parameter should *not* be treated equally in the adaptation, or tuning process. We propose to quantify the "importance" of each parameter, emphasizing preservation of important parameters during the tuning process. In the discriminative modeling setting, Kirkpatrick et al. [17] propose Elastic Weight Consolidation (EWC), which evaluates the importance of each parameter by estimating its Fisher Information relative to the objective likelihood. A key difference is in the generative setting, the training objective is not fixed. Nonetheless, we demonstrate that the Fisher Information can be estimated from a proxy objective (a frozen discriminator) and are able to generate high-quality results of different target domains, even with extremely few examples ($\leq$10).

In addition, we consider there will always be an inherent trade-off between preserving information from the source and adapting to the target domain. We conduct an in-depth analysis on the performance of our method, with respect to important factors, such as the number of target examples and the dissimilarity between the source and target domain.

The main contributions of this work are summarized as follows:

- We propose to adapt a pretrained generative model to a new target domain without introducing additional parameters, producing diverse generations even with limited data.

- We demonstrate the effectiveness of the proposed method in artistic domains, where practical use cases often have limited data.

- We evaluate our method on several cross-domain source/target pairs, in contrast to previous methods which mostly focus on the photo domain.

## 2  Related Work

**Few-shot learning.** Few-shot learning [18] is first explored in discriminative tasks where the target class contains limited labelled instances, known as few-shot image classification. It attracts considerable attention and a number of work have been done to improve the generalization performance on the target class while preventing the model from being over-fitted to the few examples. Several representative schemes include metric learning methods [39, 37], meta-learning methods [5, 27], and dynamically weight prediction methods [7]. For the target class in few-shot classification, the term *few* refers to few labels, which means there can be plenty of unlabelled images. This also leads to some semi-supervised learning methods [19]. However, in few-shot image generation, we assume that there are only a few images. No other information about the target domain is available. Another

big difference with classification is that the generation aims at generating diverse results while the classification only targets on predicting a consistent semantic label.

Most recently, a few works [28, 41] shift the attention to few-shot from discriminative to generative tasks, especially based on GANs. Some works focus on few-shot density estimation based on matching networks [1], sequential generative models [32], or autoregressive models [31] but they are limited to generating simple patterns and low resolution results. The work of [28, 41] showed first promising high resolution results on complex natural images given the recent success in high-quality GAN training. Both start from fine-tuning a pretrained GAN model and add additional parameters in some parts of the original network for learning. Such a pipeline involves many tedious manual designs and as we show later, works less effectively in extremely low-data cases. Though it preserves the ability of generating the source domain data, our goal focuses more on generating the target domain data while preserving the diversity of the source domain.

**Style transfer.** An alternative approach to generate more data of the target domain is to apply existing style transfer techniques to transfer the style of given examples (e.g., emoji style) to the abundant data of the source domain (e.g., face). There are mainly two types of style transfer methods, i.e., example-based and domain-based. Example-based schemes work with only one style example but have limitations, such as requiring alignment [9] or only transferring the color and texture [6, 12, 20, 21]. However, the style of a single example cannot fully represent a consistent style of a domain and the style of target domain sometimes is more than just color and texture, but includes the higher-level geometric shape for example. Meanwhile, domain translation methods [13, 49, 50, 38] require plenty of data for both source and target domain and thus are not directly applicable to the few-shot task. Recently some few-shot domain-based transfer work [23, 45] are proposed to solve the low-data issue in the target domain. However, their methods construct multiple source domains with style labels to either perform meta-learning or learn to disentangle the content and style representations. Different from their work, we assume only a single source domain is given.

**Continuous learning.** Since we aim at adapting a GAN model pretrained on the source domain to the target domain, this is naturally a process of sequentially learning two tasks and thus related to continuous learning. Continuous learning mainly deals with the "catastrophic forgetting" phenomenon, i.e., learning consecutive tasks without forgetting how to perform previously trained tasks. Most previous efforts are done for classification tasks, including distillation-based [22], memory-based [25], attention-based [36], and regularization-based [17, 46] methods. Based on that, several recent work [35, 47, 43] extend these to the generative domain, i.e., learning different distributions sequentially without forgetting. However, all sequential tasks learned in those work are assumed to contain enough data. The biggest difference of our focus is that for the target domain there are only a few examples. It is therefore necessary to distill knowledge learned from the previous source domain. It is also noted that after the adaptation, we are no longer able to generate the data in source domain. What we are not trying to forget here is the diversity in source domain so that we could combine it with the style from the limited data of target domain to generate more diverse results.

## 3 Proposed Method

The goal of our approach is to adapt the weights of a generative model pretrained on the source domain to the target domain with a few examples only. A direct adaptation without any regularization on the changes of weights results in over-fitting (i.e., re-generating the given examples), because the number of parameters is significantly larger than the number of examples. Therefore, the remaining questions are (i) which weights are more important to preserve or have more freedom to change and (ii) how to quantify such an importance factor so that we could regularize them via a loss function. Below we present the details of our understanding on the importance of different weights and proposed adaptation approach.

### 3.1 Rate of changes on weights

We first assume there is an abundant data in the target domain to learn a decent generative model and expect to get some inspirations on what good weights look like. With a large number of examples available, both training from scratch and fine-tuning from a pretrained model could lead to a good generative model for the target domain, while the fine-tuning simply gets faster convergence [42]. However, for the few-shot scenario, while it is unlikely to train the model from scratch with a few

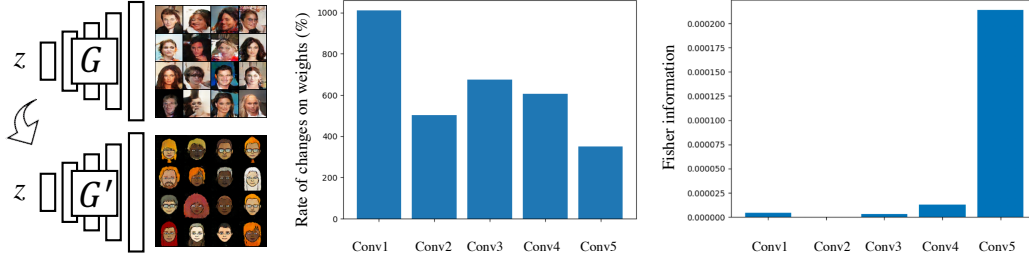

Figure 2: Analysis on weights of generative models. Left: Adapting the pretrained face model to emoji where there are abundant emoji data. Middle: The rate of changes on weights (%) at different layers between $G$ and $G'$. Right: The average Fisher information of weights at different layers in $G$.

data, we choose to analyze the fine-tuning model to identify the trend of weight changes because there are more correspondences under the same way of learning. We analyze the rate of change of the generator weights between the source and the target-adapted model. For example, it is interesting to know which weights change significantly when switching to learning another distribution. We select real faces as the source domain using the CelebA dataset [24] ($\sim$200k images). For the target domain, we use emoji faces that depict stylized human-like heads. We use the Bitmoji API [11] to collect $\sim$80k emoji images. We design a five-layer DCGAN [30] network (denoted as the generator $G$, associated with a discriminator $D$) in Figure 2 (left). We first pretrain a generative model on faces and then fine-tune it on the emoji domain, both using the following adversarial loss [8]:

$$L_{adv} = \min_{G} \max_{D} \; \mathcal{E}_{x \sim p_{data}(x)}[\log D(x)] + \mathcal{E}_{z \sim p_z(z)}[\log(1 - D(G(z)))], \qquad (1)$$

where $p_{data}(x)$ and $p_z(z)$ represent the distributions of noise variables $z$ and real data $x$. Some generated examples of faces and emoji faces are shown in Figure 2 (left).

Given a pretrained $G$ and the adapted $G'$, we compute the *average* change rate of weights at each convolution layer (here, we omit the bias and other parameters in the normalization layers): $\Delta = \frac{1}{N} \sum_i \frac{|\theta'_i - \theta_i|}{|\theta_i|}$ where $N$ is number of parameters, $\theta_i$ and $\theta'_i$ is the $i$-th parameter in model $G$ and $G'$. From the results shown in Figure 2 (middle), we observe that the weights in the last layer of the network change the least on average compared to other early layers. Similar observations are also found in other GAN architectures (e.g., LapGAN [4], StyleGAN [16]) using other source-target domain pairs. This implies that if we do the adaptation with a few examples, some weights in the last layer are more important and should be better preserved than those in other layers.

### 3.2 Importance measure

After confirming that weights in different layers should be regularized differently during the adaptation based the previous analysis, the next question is how to quantify or measure the importance of each weight. Recall that in mathematical statistics, the Fisher Information $F$ could tell how well we estimate the model parameters given the observations [26]. Given a pretrained generative model on the source domain, by generating a certain amount of data $X$ given the learned values of network parameters $\theta_S$, the Fisher information $F$ can be computed as:

$$F = \mathbb{E} \left[ -\frac{\partial^2}{\partial \theta_S^2} \mathcal{L}(X|\theta_S) \right], \qquad (2)$$

where $\mathcal{L}(X|\theta)$ is the log-likelihood function which is equivalent to computing the binary cross-entropy loss using the output of a discriminator. We also experiment with the reconstruction or perceptual loss [14] between the generated image and itself as the log-likelihood and obtain similar values for $F$. For simplicity, we use the output of the discriminator and show the average $F$ of weights at different layers in the $G$ model trained on real faces in Figure 2 (right). We notice that the weights in the last layer have much higher $F$ than those in other layers. Considering our previous observation on the rate of change of weights in Figure 2 (middle), we could directly use $F$ as an importance measure for weights and add a regularization loss to penalize the weight change during

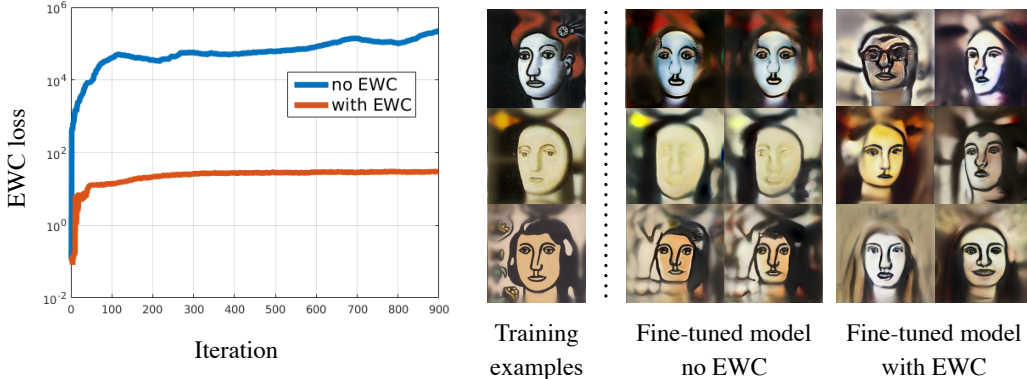

| Training examples | Fine-tuned model no EWC | Fine-tuned model with EWC |

Figure 3: The effectiveness of EWC loss. On the right, we show an example of 10-shot (3 examples are shown) generation results for an artistic target domain called Fernand Léger [44], adapted from the real face domain (all images are of size 256×256). Note that adding the EWC loss preserves the variation from source, and prevents the model from fitting the three examples exactly.

the adaptation to the target domain as follows:

$$L_{adapt} = L_{adv} + \lambda \sum_i F_i (\theta_i - \theta_{S,i})^2, \tag{3}$$

where $\theta_S$ represents the weights learned from the source domain, $i$ is the index of each parameter of the model, and $\lambda$ is the regularization weight to balance different losses. The second term in Equation (3) was first proposed in [17] for the classification task and called the Elastic Weight Consolidation (EWC) loss. While the work of [17] uses the EWC loss to avoid forgetting how to classify old classes after learning new classes and there is sufficient data for all classes, here we want to demonstrate its effectiveness in the few-shot generative setting. Without this regularization to preserve the diversity from the source, the model adaptation with a few examples in the target domain will quickly result in over-fitting, which manifests as re-generating almost-replicas of only the given target examples.

To demonstrate the effectiveness of regularization during target adaptation, we specifically ablate the second term (i.e., the EWC loss) in Equation (3). The blue curve in Figure 3 (left) shows how fast the weights are changing without any regularization. Here we compute the EWC loss for visualization but do not use it, by setting $\lambda = 0$. It clearly illustrates that the weights rapidly deviate from the original weight in just a few hundreds of iterations. We show the adapted results without EWC in the second column of Figure 3 (right), which is close to re-generating some of the 10 given examples on top and indicates that over-fitting is happening. In contrast, the orange curve in Figure 3 (left) shows that with the regularization, the weights change slowly in the beginning which also results in the increase of EWC loss, but gradually saturates. From the comparison of loss values in Figure 3 (left), we learn that adapting a few new examples only should not alter the original weight too much so that the information (e.g., diversity) from the source domain could be preserved. Adapted results with EWC in the third column of Figure 3 (right) also validate this, as our method generates new examples of the target domain.

## 4 Experimental Results

In this section, we first discuss the experimental settings. We then present qualitative and quantitative comparisons between the proposed method and several competing methods. Finally, we analyze the performance of our method with respect to some important factors such as the number of examples.

**Dataset.** We choose two objects of interests for generation, i.e., the face and landscape. We are trying to adapt the generation from real to artistic ones for those objects (e.g., real to artistic faces). We use the FFHQ dataset [16] as the source for real faces and several other face databases as the target: emoji faces from the Bitmoji API [11]; animal faces from the AFHQ dataset [3] and portrait paintings from the Artistic-Faces dataset [44]. We use 10 cat and dog images from the, much larger, AFHQ dataset. The Artistic-Faces dataset contains artistic portraits of 16 different artists and there are only 10 images per artist available. For the landscape, we use the CLP dataset [29] that contains thousands of landscape photos as the source and 10 pencil landscape drawings as the target.

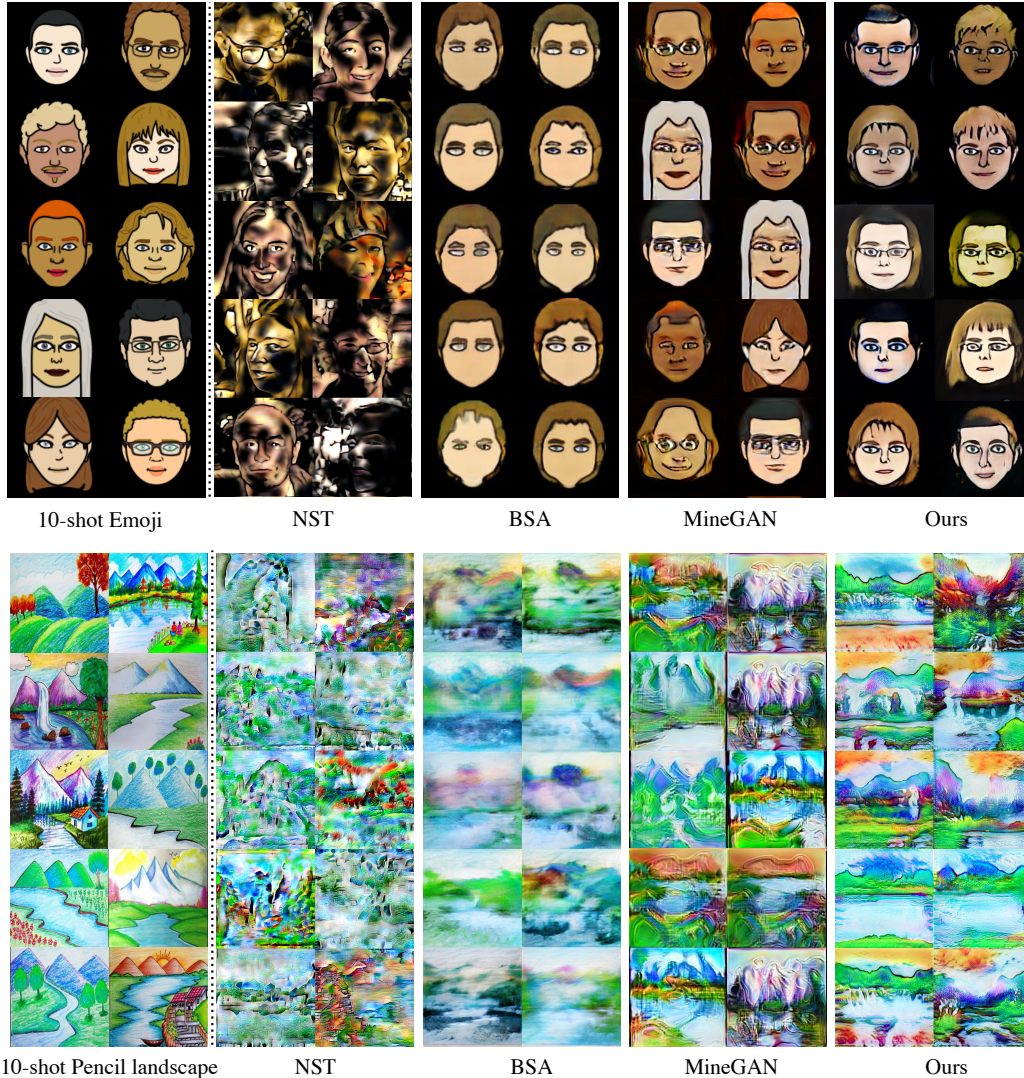

| 10-shot Emoji | NST | BSA | MineGAN | Ours |

| 10-shot Pencil landscape | NST | BSA | MineGAN | Ours |

Figure 4: Visual comparisons of different methods for few-shot generation. Top: **FFHQ → Emoji**; Bottom: **Natural landscape → Pencil landscape**. In each example, Left: 10-shot training examples; Right: 10 generated results by each method (all images are of size 256×256).

**Evaluated methods.** We evaluate our method against three others: NST [6], BSA [28], and Mine-GAN [41]. The Neural Style (NST) work of [6] represents a family of neural style transfer methods as adaptation to a new domain can be also regarded as a style transfer task. As it is an example-based method, we randomly select one image from the small amount of target examples to stylize an image sampled from the source data. Both [28] and [41] also focus on adapting models from a source to a target domain but they introduce additional parameters. The BSA method [28] is adding new batch norm layers into the original BigGAN generator [2] and learning new parameters only during the adaptation. The MineGAN approach [41] adds a small mining network $M$ in front of the original Progressive GAN generator [15] and proposes a two-stage fine-tuning strategy (i.e., fine-tune $M$ first and then fine-tune $M$ jointly with the generator). For all experiments of our method in this work, we use the StyleGAN [16] framework.

## 4.1 Qualitative results

Figure 4 shows a visual comparison between different methods. The top one is adapted from the real face domain and the bottom one is adapted from the natural landscape domain. NST mostly transfers

Table 1: Quantitative comparisons between different few-shot generation methods. For FID and LPIPS, each result is in the form of {mean ± standard error}. For the user study, the result (i.e., fooling rate) is in the form of {probability ± confidence interval} with the 95% confidence level.

|  | Source | NST [6] | BSA [28] | MineGAN [41] | Ours |
|---|---|---|---|---|---|
| FID↓ | $42.58 \pm 1.76$ | $204.16 \pm 9.28$ | $105.56 \pm 5.79$ | $86.44 \pm 4.38$ | $\mathbf{74.87 \pm 3.75}$ |
| LPIPS↑ | $0.46 \pm 0.03$ | $\mathbf{0.57 \pm 0.01}$ | $0.15 \pm 0.01$ | $0.21 \pm 0.02$ | $0.40 \pm 0.02$ |
| User (%)↑ | – | $15.28 \pm 5.58$ | $4.86 \pm 3.33$ | $30.56 \pm 7.14$ | $\mathbf{47.92 \pm 7.38}$ |

Table 2: Quantitative comparisons between different few-shot generation methods with respect to the number of shots (FID↓).

| Number of shots | NST [6] | BSA [28] | MineGAN [41] | Ours |
|---|---|---|---|---|
| 1 | $212.23 \pm 9.77$ | $102.34 \pm 5.70$ | $102.57 \pm 4.76$ | $\mathbf{84.36 \pm 3.91}$ |
| 10 | $204.16 \pm 9.28$ | $105.56 \pm 5.79$ | $86.44 \pm 4.38$ | $\mathbf{74.87 \pm 3.75}$ |
| 100 | $199.52 \pm 9.02$ | $110.24 \pm 5.87$ | $76.23 \pm 4.05$ | $\mathbf{67.55 \pm 3.48}$ |
| 1,000 | $196.43 \pm 8.83$ | $119.31 \pm 5.96$ | $69.20 \pm 3.59$ | $\mathbf{62.40 \pm 3.12}$ |
| 10,000 | $194.88 \pm 8.71$ | $131.20 \pm 6.04$ | $58.69 \pm 3.14$ | $\mathbf{55.74 \pm 2.88}$ |

global color and texture of the target examples. The transferred results are relatively cluttered and do not capture higher-level characteristics (e.g., geometric shape) of the target style. The results of BSA look blurry at the first glance due to their reconstruction scheme. Moreover, it also shows the problem of mode collapse by generating visually similar results. MineGAN is less effective in generating more diverse examples. It is easy to spot re-generation of given examples in the sampled results, which implies that MineGAN tends to overfit under the few-shot setting. In contrast, our method generates results that are more faithful to the style of the few given examples, while exhibiting a good amount of diversity.

## 4.2 Quantitative comparisons

While we only use a few examples from the target domain to perform the adaptation, we divide the quantitative study into two parts, based on whether the target domain contains abundant data for evaluation. If the target domain originally has a lot of real data, we select the commonly used Fréchet Inception Distance (FID) [10] which measures the quality of generated images obtained by the adapted model. For target domains that have only a few examples available, FID is not a good metric for measuring the generation quality. Therefore, we conduct user studies to evaluate how realistic our generated results are compared with real examples. At the top of the user study page, we show a few real examples from the artistic domain as reference. Below, we show one real example outside of the reference set and one generated result side by side. Each user is asked to select which one is *generated* and do 10 rounds of selection. The user study is conducted per method and we use the Amazon Mechanical Turk (AMT) platform to collect 300 votes from users in each study. A better method should fool users more easily to make a wrong decision. For measuring diversity, we use the LPIPS metric [48] to measure the similarity among results, i.e. the distance between a number of pairs of randomly generated images.

Table 1 shows the quantitative comparison results between different methods with 10-shot generation. We first evaluate the pretrained models on the source domain and show their results as the reference. The results in the first row of Table 1 clearly show that our method achieves the lowest FID score compared with other three schemes, which is consistent with the better results obtained by our method shown in Figure 4. For the diversity evaluation results in the second row, it is noted that NST achieves the highest LPIPS score which is, however, due to the cluttered transferred artifacts in their results. Therefore it is more meaningful to compare with [28, 41] and the comparisons show that our method is able to generate more diverse results. The third row presents the user study results when evaluating the realism of generated results for target domains with 10 examples only. Each result represents the probability of being fooled (i.e., fooling rate) by selecting the wrong answer. The higher percentage value means the results obtained by a certain method are more realistic so that they tend to fool the users between real and generated examples. Our method obtains the highest fooling rate 47.92%, which means our results are more indistinguishable from real images.

Table 3: Analysis on the performance with respect to the regularization weight $\lambda$.

| $\lambda\,(\times 5)$ | $10^6$ | $10^7$ | $10^8$ | $10^9$ | $10^{10}$ |
|---|---|---|---|---|---|
| FID $\downarrow$ | 78.21 | 75.62 | 74.87 | 77.79 | 80.21 |
| LPIPS $\uparrow$ | 0.383 | 0.391 | 0.405 | 0.412 | 0.419 |

Table 4: Analysis on the performance with respect to dissimilarity between source and target.

| | Female | Emoji | Cat | Landscape |
|---|---|---|---|---|
| FID $\downarrow$ | 67.08 | 76.25 | 107.67 | 164.83 |
| LPIPS $\uparrow$ | 0.459 | 0.596 | 0.651 | 0.735 |

1-shot Emoji 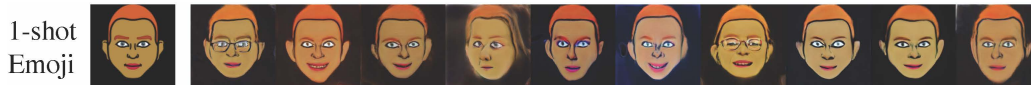

Figure 5: 1-shot (leftmost) adaptive generation from the real face domain FFHQ to the emoji domain. Our method generates variations of the given example.

## 4.3 Discussion

**Number of shots.** As a key factor in few-shot generation, the number of examples in the target domain plays an important role in affecting the performance. Generally, provided with more examples, we expect better adaptation performance of the proposed method. If there is enough data, we could directly learn the distribution from scratch instead of doing the adaptation. We additionally evaluate our method against existing approaches on other shots in Table 2. The performance of NST will not be improved obviously because the style transfer method itself cannot capture the target style well, regardless of number of examples. An observation from the BSA [28] work is that their performance drops when increasing the number of examples. This is due to their strategy of reconstructing training examples and more examples result in lower-quality reconstructions. The MineGAN and our work behave similarly as both are distribution learning based methods. The more data, the better performance and the closer these two methods are. However, our method clearly exhibits more obvious advantage over MineGAN in extremely few data scenarios (e.g., $\leq 10$).

We would like to highlight the special case of 1-shot. When the target domain contains one example only, it is unlikely to expect big amount of diversity in results no matter how strongly we regularize the weight. What we observe is that the adapted model is generating variations of the given example. Figure 5 shows an example of 1-shot adaptation from the FFHQ source to the emoji target domain. Though all generated results are with brown-like skin and red-like hair, they present meaningful differences between each other, e.g., the glass, smile, tooth, pose, and gender. Our strategy is still effective in inheriting the information from the source domain.

**Regularization weight $\lambda$.** The parameter $\lambda$ in Equation (3) controls the power of regularization term added during the adaptation. We show its effect on the performance in Table 3 (10-shot). A larger value of $\lambda$ would preserve too many details of the source, which hinders the adaptation to the target domain but preserves more diversity. A smaller value of $\lambda$ gives too much freedom on the changes of weight, which may result in the over-fitting to the target domain and reduce the diversity. This represents the unavoidable trade-off we achieved between inheriting from the source and adapting to the target. We empirically set $\lambda = 5 \times 10^8$ in all our experiments. In addition, we find that (i) if the source and target are more similar (e.g., from the male face to female face), select a larger $\lambda$ to constrain the weight changes because a minor change might be enough for the adaptation, and (ii) if more target data is given, select a smaller $\lambda$. An extreme case is that there is no need to do any regularization (i.e., $\lambda = 0$) if there is abundant data available in target domain.

**Dissimilarity between source and target.** An imperative assumption before doing the adaptation in our method is that the source and target domain describe the same object (e.g., real faces and emoji faces) so that they share latent factors (e.g., poses) except for their distinct difference in appearance. It does not make too much sense if we want to adapt a face model to the landscape domain. Therefore one conjecture is that the performance of the propose method will decrease when the source and target domain become more dissimilar. To validate this, we select the FFHQ [16] face dataset as the source domain and several target domains for adaptation according to their dissimilarity with FFHQ: the CelebA-Female face [24], the emoji face [11], the cat face [3], and the color pencil landscape [29]. The dissimilarity between images in two domains is measured using the LPIPS [48] metric. We conduct a 10-shot adaptation for each target domain and evaluate the results with the FID score (those

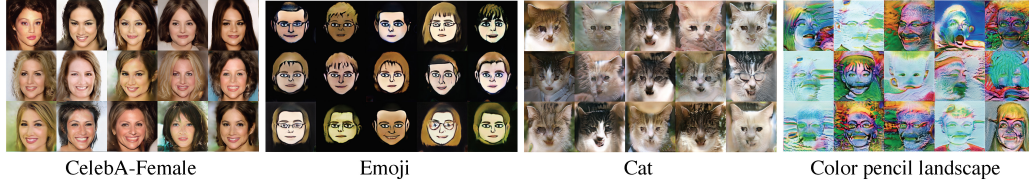

| CelebA-Female | Emoji | Cat | Color pencil landscape |

Figure 6: 10-shot generation results of four different target domains all adapted from the same source domain: real faces. From left to right, when the target domain is more and more dissimilar with the real face domain, the adapted results become more and more unrealistic.

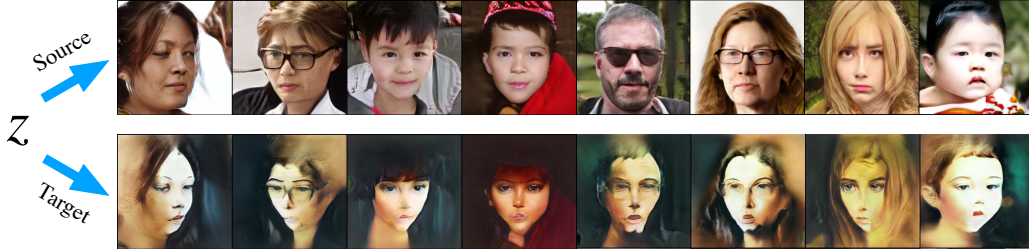

Figure 7: Generations of source and target domain by feeding the same latent code into the source (FFHQ) and adapted target (the Moïse Kisling faces in Figure 1) model.

target domains all originally contain a lot of data for evaluation). Results in Table 4 clearly validate our previous conjecture.

We show examples of generated new results of those four target domains in Figure 6. On the leftmost, the results of CelebA-Female face are the most realistic and diverse as this target domain is closest to the source domain FFHQ. Simply changing some low-level facial textures could easily achieve the altering of gender. On the rightmost, the results of color pencil landscape domain adapted from FFHQ obviously do not make much sense as we could still observe the silhouette of face preserved. Adding the EWC regularization is not sufficient to change the semantic shape from face to landscape. This implies that to make the adaptation work more successfully, it is better to consider selecting a similar source domain to do the pretraining.

**Correspondence.** To demonstrate whether the diversity is preserved from the source during the adaptation, one straightforward way is to visualize if certain level of correspondence exists between the generated results of source and target domain by feeding the same latent code into the source and adapted target model. As shown in Figure 7, while the adaptation renders new appearance of target domain, other attributes such as the pose, glass and hairstyle, are well inherited and preserved from the source domain. Given that collecting real paired data is often labor intensive, one promising aspect of our method is that we can obtain unlimited number of synthetic paired data by leveraging the correspondence between the source and target model.

## 5 Conclusion

In this work, we focus on the challenging task of unconditional image generation in low-data regime. Given a few examples only in the target domain, we adapt a pretrained generative model learned on the source domain with abundant data to generate more data of the target domain. Inspired by the continuous learning, we analyze the weight importance and quantify such a factor in order to selectively regularize the weight changes during the adaptation. In this way, we achieve the inheritance of diversity from the source domain as well as the adapting of new appearance from the target domain, and avoid the over-fitting issue known to easily happen when data is limited. The proposed method is simple and effective, and may shed light on more future understandings of the learned parameters. We demonstrate the efficacy of the proposed method on various domains and show that it performs favorably against existing methods. The new generated data could expand the variety of the domain that is originally scarce in data and consequently facilitate many downstream image synthesis tasks.

## Broader Impact

**AI for creativity.** The motivation of this work is to expand the amount of data in domains where originally there is limited data available. It is especially useful for artistic domains where manually making a creation takes a lot of work and time. With the generated data, many existing AI-based image synthesis pipelines could be facilitated with the large-scale training. We believe more creative applications could benefit from our work in terms of constructing the indispensable dataset.

**Detectability.** While our use cases in this work are geared towards creative applications, a concern is the generated imagery can be used for the purpose of deception. A potential mitigation is if generated imagery can be reliably detected; there are recent efforts [40, 33] made in this area. The latest work by Wang et al. [40] shows that a classifier trained on images generated by one method, could generalize to others, despite different architectural components or loss functions. As we are using an architecture with many shared components, we expect this generalization ability to hold. We conduct a small study, using the Blur+Jpeg(0.5) model from [40] on our Cat and CelebA-Female datasets. We find the model achieves 94.9% and 99.6% average precision (AP), respectively, for classifying generated images. This indicates our method is similarly detectable to already existing CNN-generated methods.

While these results are strong, they are not 100%. Furthermore, performance can degrade as the images are degraded in real use cases (e.g., compressed, re-scanned). The issue of content authenticity remains a significant challenge, likely requiring multiple layers of solutions, from technical (such as this detector from [40]), to social, to regulatory.

## Acknowledgments and Disclosure of Funding

We thank anonymous reviewers for their constructive and valuable feedback on the early submission. This work is not supported by any third-party funding.

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
