[Supplementary Material]

# Few-shot Image Generation with Elastic Weight Consolidation Supplementary Material

**Yijun Li**    **Richard Zhang**    **Jingwan Lu**    **Eli Shechtman**

Adobe Research
{yijli, rizhang, jlu, elishe}@adobe.com

## 1   Overview

In this supplementary material, we present more few-shot generation results evaluated extensively with different artistic domains where there are only a few examples available in practical. The goal is to illustrate the effectiveness of the proposed method in generating diverse high-quality results without being over-fitted to the few given examples.

Figure 1 shows the generations of source and target domain by feeding the **same** latent code into the source and adapted model. It clearly tells that while the adaptation renders new appearance of target domain, other attributes such as the pose, glass and hairstyle, are well inherited and preserved from the source domain.

Figure 2~18 show more generation results ($256\times256$) for different target domains. For each target domain, we only use 10 examples for the adaptation and present 100 new results.

Figure 19~20 show higher-resolution generation results where both the source and adapted model are trained with higher-resolution images (i.e., 512 and 1024).

Figure 21 shows the user interface of our designed user study.

10-shot examples of target domain Moise Kisling

Generations of source domain FFHQ

Generations of target domain Moise Kisling

Figure 1: Generations of source and target domain under the **same** input noise. Images on the left and right panel at the same position correspond to the same latent code.

10-shot examples

Adapted generations

Figure 2: Diverse generated results for the target domain (10 examples only): Amedeo Modigliani, adapted from the source: FFHQ (256×256).

10-shot examples

Adapted generations

Figure 3: Diverse generated results for the target domain (10 examples only): Comics, adapted from the source: FFHQ (256×256).

10-shot examples

Adapted generations

Figure 4: Diverse generated results for the target domain (10 examples only): Egon Schiele, adapted from the source: FFHQ (256×256).

10-shot examples

Adapted generations

Figure 5: Diverse generated results for the target domain (10 examples only): Fernand Léger, adapted from the source: FFHQ (256×256).

10-shot examples

Adapted generations

Figure 6: Diverse generated results for the target domain (10 examples only): Henri Matisse, adapted from the source: FFHQ (256×256).

10-shot examples

Adapted generations

Figure 7: Diverse generated results for the target domain (10 examples only): Hindu Gods, adapted from the source: FFHQ (256×256).

10-shot examples

Adapted generations

Figure 8: Diverse generated results for the target domain (10 examples only): Israhel van Meckenem, adapted from the source: FFHQ (256×256).

10-shot examples

Adapted generations

Figure 9: Diverse generated results for the target domain (10 examples only): John Bratby, adapted from the source: FFHQ (256×256).

10-shot examples

Adapted generations

Figure 10: Diverse generated results for the target domain (10 examples only): Marc Chagall, adapted from the source: FFHQ (256×256).

10-shot examples

Adapted generations

Figure 11: Diverse generated results for the target domain (10 examples only): Moïse Kisling, adapted from the source: FFHQ (256×256).

10-shot examples

Adapted generations

Figure 12: Diverse generated results for the target domain (10 examples only): Otto Dix, adapted from the source: FFHQ (256×256).

10-shot examples

Adapted generations

Figure 13: Diverse generated results for the target domain (10 examples only): Pablo Picasso, adapted from the source: FFHQ (256×256).

10-shot examples

Adapted generations

Figure 14: Diverse generated results for the target domain (10 examples only): Raphael, adapted from the source: FFHQ (256×256).

10-shot examples

Adapted generations

Figure 15: Diverse generated results for the target domain (10 examples only): Roy Lichtenstein, adapted from the source: FFHQ (256×256).

10-shot examples

Adapted generations

Figure 16: Diverse generated results for the target domain (10 examples only): Utagawa Kunisada, adapted from the source: FFHQ (256×256).

10-shot examples

Adapted generations

Figure 17: Diverse generated results for the target domain (10 examples only): Vincent van Gogh, adapted from the source: FFHQ (256×256).

10-shot examples

Adapted generations

Figure 18: Diverse generated results for the target domain (10 examples only): Color pencil landscape, adapted from the source: Natural landscape (256×256).

10-shot examples

Adapted generations (512×512)

Adapted generations (1024×1024)

Figure 19: Diverse generated results for the target domain (10 examples only): Moïse Kisling, adapted from the source: FFHQ.

10-shot examples

Adapted generations (512×512)

Adapted generations (1024×1024)

Figure 20: Diverse generated results for the target domain (10 examples only): John Bratby, adapted from the source: FFHQ.

Figure 21: User interface of our user study.