[Reviews · NeurIPS 2020]

Review 1

Summary and Contributions: This paper presents a methodology to train GAN on few-shot learning paradigm for a domain where limited training examples are available. To achieve this, authors propose to fine-tune the model parameters trained on a domain with abundant training examples by setting variable layerwise learning rate. To identify the suitable learning rate, authors employ Fisher Information (FI). Also, adopts the elastic consolidation loss as a regulariser which has been successfully applied before the training discriminative model to fine-tune the model. The proposed method is evaluated on multiple source-target domain pairs. Mostly qualitative and also quantitative comparisons are made with some of the existing arts.

Strengths: Few-shot paradigm for training a generative model is an interesting research problem. Extensive qualitative comparisons are made. Qualitative results seem better than existing art on the compared set up. Quantitative measurements also show the efficacy of the proposed method with the existing art. The paper is generally well written. I would consider the novelty of the method is moderate as the regulariser has been previously applied for training discriminative model.

Weaknesses: To identify the trend of layerwise fine-tuning rate, authors took CelebA (200K) images as the source domain and Bitmoji (80K) as a target domain (line 120-122). However, authors have taken downstream scenario as one having very limited training examples ( from 1 to 10). What guarantees that the earlier scenario generalises with the latter scenario? After rebuttal, I still feel how the fine-tuned version will be better when you already have such a large data set. Again referring to the above comment, if we already have examples up to 80K, does the model just fine-tune or learns the parameters specific for the target domain? Is there any guarantee that the fine-tuned version is better than the one trained from scratch?

Correctness: FID as a metric to evaluate the quality of synthetic data is reasonable. However, authors compared the existing arts with at a single point (10-shot) only (Table 1). Seeing only this comparison, it is very hard to draw a concrete conclusion. Hence, it is important to compare on other shot as well (Table 2). On rebuttal, the results on other shots were reported and it looks convincing. Hence, I am up voting my rating.

Clarity: Yes, this paper is generally a well written paper.

Relation to Prior Work: Generally, related works are discussed well. Need to discuss the following reference too: Zakharov, Egor, et al. "Few-shot adversarial learning of realistic neural talking head models." Proceedings of the IEEE International Conference on Computer Vision. 2019.

Reproducibility: Yes

Additional Feedback:


Review 2

Summary and Contributions: In this paper, authors propose a novel approach to few-shot image generation that utilizes regularized finetuning according to the importance of the parameters. the paper includes extensive and convincing qualitative and quantitative evaluation of the proposed method and a thorough analysis of the impact of the amount of target examples and dissimilarity of source and target datasets on the quality and diversity of image generation.

Strengths: The proposed method is sufficiently novel an reasonable. Authors provided either empirical evaluation results or appropriate references to almost every claim they make in the paper. The experimental setup is thoroughly thought through and the appropriate metrics are used to prove their hypothesis. The qualitative results indicate clear advantage of the proposed solution. Finally, the paper is easy to read and well organized.

Weaknesses: The quality of the paper is already great, but there are a few comments. 1. In equation 3 (page 4), it is not clear whether you compute F of the generated source or target data. Also, I don't quite understand why the FI is computed for the difference between the pretrained and finetuned parameters, and not just for the pretrained parameters. Finally, I assume i in this equation is the layer index, but this should be clearly stated. Update: In the rebuttal, the authors kindly explained that the F is computed for each individual parameter in the network rather than for the entire layer. I suggest adding this clarification to the main paper as well. 2. In Figure 3, it would be very illustrative to show how each layer affects the generation. You can do this by regularizing everything but the layer and look at the generation result. This would further convince the reader that some layers are more important than others for the diversity preservation. 3. In Table 3, it is unclear from the caption what source dataset is used. Update: Please add some information about the source dataset in the caption. The image and table captions should contain all the necessary information about the experiment so that the reader doesn't have to search for in the main text. Apart from that, great job!

Correctness: Apart from the claim that the last layers play a more significant role in diversity preservation (which was clarified in the rebuttal), all the main claims of the paper are intuitive or proved empirically.

Clarity: The paper is very well-written.

Relation to Prior Work: The prior work is covered well, although it would be better if the baselines were described in more detail in the related work section.

Reproducibility: No

Additional Feedback:


Review 3

Summary and Contributions: The paper proposes to regularize the network parameters changes during the source to target adaptation progress for few-shot image generations. The method has been demonstrated to effectively preserve the “information” of the source dataset, while fitting the target.

Strengths: 1. The idea of quantifying the “importance”of each parameter (deep networks' parameters) for adaptation tuning is interesting and novel. It effectively transfers the knowledge to target datasets for few-shot image generations. 2. The qualitative and quantitative comparisons are well analyzed and can demonstrate the effectiveness of the proposed method. 3. The paper is well written and easy to follow.

Weaknesses: Overall the idea introduced in the paper is novel and inspiring, though the regularization terms in Eq. 3 was proposed in [17]. The reviewer has some concerns about the experiments. 1. The experiments are mainly performed using different face datasets in the image size of 256*256. Can the approach be used for high-resolution settings or synthesis where the images contain many structured details? (e.g., urban driven images and indoor images)? 2. Since a key contribution is using Eq. 2 (F_i) to weighting the regularization terms (\theta_i - \theta_{S, i})^. Some straightforward options (baselines) should be compared. For example, give some fixed weights for each convolutional layer according to Figure. 2 (middle). Besides, How about removing F_i?

Correctness: Yes.

Clarity: Yes.

Relation to Prior Work: Yes.

Reproducibility: Yes

Additional Feedback: The rebuttal addresses some of my concerns. The main contribution of the paper is their interesting idea, i.e., quantifying the “importance”of each parameter (deep networks' parameters) for adaptation tuning. The regularization term is not a major contribution since it has been studied in [17]. I will keep my borderline positive rating.


Review 4

Summary and Contributions: This paper proposes a method to achieve few-shot image generation by regularizing the changes of the weights during adaptation to best preserve the information of source dataset. The effectiveness of the proposed algorithm is demonstrated by generating high-quality results of different target domains.

Strengths: The proposed self-adaption technique producing diverse generations with limited data. The effectiveness of the proposed method is demonstrated in artistic domains and several cross-domain source/target pairs, in contrast to previous methods which mostly focus on the photo domain. The proposed method is evaluated on several cross-domain source/target pairs in contrast to previous methods which mainly focus on the photo domain.

Weaknesses: The main contribution of this paper is the weight regularization of the changes of the weights during this adaptation to best preserve the information. Thus, the technique contribution of this work is limited. In line 132, it is claimed that the later layers in generators are mainly responsible for synthesizing low-level features, which are more likely to be shared across domains. It is hard to understand, because semantic features in high level tend to be shared across domains. Missing some analysis of regularization parameters. It is claimed that the parameters is determined empirically, while how about the performance with different regularization parameters?

Correctness: Yes

Clarity: Yes

Relation to Prior Work: Yes

Reproducibility: Yes

Additional Feedback: Seeing the rebuttal and other reviews, I decide to raise my score.

[Author Response · NeurIPS 2020]

We thank the recognition from reviewers on the value of research problem, novelty of the proposed method and our
results. We address major raised concerns below. One missing reference (R1) will be added in the revised version.

**R1: fine-tuned or trained from scratch.** The work of [42] has shown that given enough data, training from scratch
or fine-tuning from a pretrained model could both learn a good generative model for the target domain, while the
fine-tuning simply gets faster convergence. As stated on L2-4, for the few-shot scenario, while it is unlikely to train the
model from scratch with a few data, we design our pipeline in the fine-tuning fashion. This is why we identify the trend
from the fine-tuned model because there are more correspondences on weights changes under the same way of learning.

**R1: FID on other shots.** We additionally evaluate our method against existing approaches on other shots in the
following table. The performance of NST will not be improved obviously because the style transfer method cannot
capture the target style well (L200-202). As stated on L244-246, the BSA's performance drops when increasing the
number of shots. The MineGAN and our work behave similarly as both are distribution learning based methods. The
more data, the better performance and the closer these two methods are.

Table 1: Quantitative comparisons between different few-shot generation methods (FID↓).

| Number of shots | NST [6] | BSA [28] | MineGAN [41] | Ours |
|---|---|---|---|---|
| 1 | $212.23 \pm 9.77$ | $102.34 \pm 5.70$ | $102.57 \pm 4.76$ | $84.36 \pm 3.91$ |
| 10 | $204.16 \pm 9.28$ | $105.56 \pm 5.79$ | $86.44 \pm 4.38$ | $74.87 \pm 3.75$ |
| 100 | $199.52 \pm 9.02$ | $110.24 \pm 5.87$ | $76.23 \pm 4.05$ | $67.55 \pm 3.48$ |
| 1,000 | $196.43 \pm 8.83$ | $119.31 \pm 5.96$ | $69.20 \pm 3.59$ | $62.40 \pm 3.12$ |
| 10,000 | $194.88 \pm 8.71$ | $131.20 \pm 6.04$ | $58.69 \pm 3.14$ | $55.74 \pm 2.88$ |

**R2: clarification on FI.** The Fisher Information is computed for each pretrained parameter $\theta_{S,i}$ in the **source** model,
and used as an importance weight to regularize the changes of each parameter $\theta_i$ during the adaptation on target domain.
The $i$ is the index of each parameter in the model. Note that the FI is computed for each single parameter instead of
each layer. What Figure 2(right) shows is the **average** FI of all parameters at each layer for easier visualization.

**R2: effect of each layer.** Thanks for the suggestion to ablate the importance of each layer and we will include it in the
revised draft. We mainly focus on the individual parameter because the FI is computed for each parameter.

**R2: source domain in Table 3.** As stated on L261, we select the FFHQ face dataset as the source domain.

**R3: other datasets.** In addition to results ($256 \times 256$) on face datasets, we also presented higher-resolution ($512 \times 512$,
$1024 \times 1024$) face generation in Figure 19-20 of the supplementary material and the landscape generation in Figure 4 of
the paper. We think our approach can be also used for other datasets with more structures details based on two premises:
i) a decent pretrained model on the source domain and ii) some level of similarity between the source and target domain.

**R3: more baselines on $F_i$.** The baseline of removing $F_i$ (i.e., without EWC used) is shown in Figure 3 of the paper
and discussed on L161-173. We also add another baseline by giving some fixed weights (i.e., the average FI in Figure 2
(right)) for all parameters in each layer. We observe that it will not lead to obvious over-fitting and the performance is
slightly worse (FID: 77.16 vs. Ours 74.87 for 10-shot). Comparing with treating all parameters at each layer equally
with the same weight, ours by regularizing each single parameter with its own FI still works better.

**R4: limited contribution.** Technically, our work on regularizing the model's own parameters is to **avoid** introducing
additional new parameters as did in existing approaches [28,41] which involves many tedious manual designs (e.g., the
number of parameters, the position to embed those parameters). The proposed method is simple and effective, and
may shed light on more future understandings of the learned parameters. Practically, we proposed a solution to the
challenging few-shot generation problem (e.g., 10 examples) as generative models often struggle in the low-data regime.

**R4: later layers are important.** Thanks for pointing out this improper statement. We shall not conclude parameters in
the later layer are all important based on the **average** FI. It should be some important parameters with much larger FI
that results in the higher average FI. We will redesign Figure 2 with the standard deviation for better visualization.

**R4: analysis of the regularization weight.** We assume R4 is referring to $\lambda$ in Eq. (3) as this is the only parameter we
empirically set the value for. We show its effect on the performance (FID) in the following table (10-shot). A large $\lambda$
tends preserve more style of the source and a small $\lambda$ may result in the some level of over-fitting. In addition, we find
that (i) if the source and target are more similar, select a larger $\lambda$, and (ii) if more target data is given, select a smaller $\lambda$.

| $\lambda$ | $5 \times 10^6$ | $5 \times 10^7$ | $5 \times 10^8$ | $5 \times 10^9$ | $5 \times 10^{10}$ |
|---|---|---|---|---|---|
| FID ↓ | $78.21 \pm 3.80$ | $75.62 \pm 3.74$ | $\mathbf{74.87 \pm 3.75}$ | $77.79 \pm 3.78$ | $80.21 \pm 3.82$ |

[Meta-Review · NeurIPS 2020]

This paper proposes an interesting framework for generating images from few-shot data. While some reviewers were concerned about the quality of generated images, the ideas in this paper is interesting enough to justify publicaition.